# Atsttrin regulates osteoblastogenesis and osteoclastogenesis through the TNFR pathway

Kaiwen Liu [1], Zihao Wang[1], Jinbo Liu[1], Wei Zhao[1], Fei Qiao[1,2], Qiting He[3], Jie Shi[1], Qunbo Meng[1], Jianlu Wei [1✉] & Lei Cheng [1✉]

Osteoporosis is a systemic metabolic bone disorder for which inflammatory cytokines play an important role. To develop new osteoporosis treatments, strategies for improving the microenvironment for osteoblast and osteoclast balance are needed. Tumor necrosis factor-α (TNF-α) plays an important role in the initiation and development of osteoporosis. Atsttrin is an engineered protein derived from the growth factor, progranulin (PGRN). The present study investigates whether Atsttrin affects osteoclast formation and osteoblast formation. Here we show Atsttrin inhibits TNF-α-induced osteoclastogenesis and inflammation. Further mechanistic investigation indicates Atsttrin inhibits TNF-α-induced osteoclastogenesis through the TNFR1 signaling pathway. Moreover, Atsttrin rescues TNF-α-mediated inhibition of osteoblastogenesis via the TNFR1 pathway. Importantly, the present study indicates that while Atsttrin cannot directly induce osteoblastogenesis, it can significantly enhance osteoblastogenesis through TNFR2-Akt-Erk1/2 signaling. These results suggest that Atsttrin treatment could potentially be a strategy for maintaining proper bone homeostasis by regulating the osteoclast/osteoblast balance. Additionally, these results provide new insights for other bone metabolism-related diseases.

[1] Department of Orthopedic, Qilu Hospital of Shandong University, Jinan, Shandong 250012, China. [2] Department of Pediatric Orthopedic, Dalian Women and Children's Medical Center(group), Dalian, Liaoning 116012, China. [3] Department of Orthopedics, Honghui Hospital, Xian Jiaotong University, Xian, Shanxi 710054, China. ✉email: 18560089157@163.com; chenglei@email.sdu.edu.cn

Bone matrix homeostasis is a dynamically balanced system that is maintained by a balance between bone formation by osteoblasts and bone resorption by osteoclasts[1,2]. Osteoblasts are derived from mesenchymal precursor cells and are responsible for the deposition of new bone matrix to facilitate bone mineralization during bone reconstruction[3,4]. Osteoclasts, on the other hand, are giant multinucleated cells derived from the bone marrow monocyte lineage that play an important role in reabsorbing mineralized matrix[5,6]. Osteoporosis is defined as a systemic metabolic bone disease[7]. In osteoporosis, the balance between osteoclasts and osteoblasts is disturbed, resulting in increased bone resorption and/or decreased bone formation, which in turn reduces bone mass and bone microstructure destruction, eventually making bones prone to fracture[8]. Inflammatory cytokines play an important role in the occurrence and development of osteoporosis[9]. TNF-α is a member of the tumor necrosis factor-α superfamily, often at the peak of the inflammatory cascade[10–12]. This study found that the TNF-α level in women with postmenopausal osteoporosis was higher than that in women without osteoporosis[13]. TNF-α increases receptor activator of nuclear factor kappa-B ligand (RANKL) secretion in osteoblasts and bone marrow stromal cells, which can further induce osteoclast formation and enhance osteoclast activity and bone resorption[14]. It was found that the expression of TNF-α in the peripheral blood and local bone tissue of osteoporotic mice was increased in age-related and ovariectomy-induced osteoporosis mouse models, which was positively correlated with the pathological process of osteoporosis[15]. At the same time, TNF-α can also inhibit the differentiation of osteoblasts and eventually lead to the destruction of normal bone metabolism balance, resulting in bone loss[16,17].

Progranulin (PGRN) is a multifunctional and widely expressed growth factor that plays an important role in many pathophysiological processes of the human body, including anti-inflammatory processes, cartilage formation, tumorigenesis, tissue repair, and wound healing. PGRN contains seven semi-repeated cysteine-rich motifs that can bind to tumor necrosis factor receptors (TNFRs) and effectively inhibit the TNFα-mediated inflammatory response, which can play a protective role in inflammatory diseases[10,18,19]. Atsttrin is a PGRN-derived engineered protein composed of three fragments of PGRN, including half-unit granulin A, C, and F plus the linked proteins P3, P4 and P5. Atsttrin is the smallest molecule of PGRN function and still has an affinity for the TNF-α receptor[20]. Atsttrin and PGRN have similar physiological functions and can also bind to TNFR and inhibit TNF-α signaling pathways[20,21]. Moreover, Atsttrin showed longer half-lives and stronger therapeutic effects in previous studies compared to PGRN[21–23]. At the same time, we found that Atsttrin plays an important role in protecting early inflammatory arthritis in animal models and promoting cartilage repair[20,21].

However, researchers have not clearly determined whether Atsttrin plays a role in preventing the development of osteoporosis. Based on the results of previous studies, taking into account the importance of TNF-α in osteoporosis, the purpose of this study was to investigate the role of Atsttrin in regulating osteoclast and osteoblast formation as well as the potential molecular mechanism involved, providing an alternative strategy for the treatment of osteoporosis.

## Results

### Atsttrin inhibited TNF-α-induced osteoclastogenesis.
TNF-α is reported to play a predominant role in osteoclastogenesis[14,24,25]. To determine the role of Atsttrin in TNF-α- and/or RANKL-mediated osteoclastogenesis, we cultured BMDMs in the presence of RANKL and/or TNF-α for 7 days, followed by TRAP staining. As indicated in Fig. 1a, b, RANKL effectively induced osteoclast differentiation, while additional use of TNF-α significantly enhanced RANKL-induced osteoclastogenesis. Interestingly, Atsttrin significantly rescued TNF-α- enhanced osteoclast formation. In addition, we found that Atsttrin was unable to inhibit RANKL-induced osteoclast differentiation directly. Meanwhile, the results showed that TNF-α could not directly promote osteoclast differentiation in the absence of RANKL. To further confirm this finding, we cultured BMDMs with RANKL (100 ng/ml) in the absence or presence of TNF-α (10 ng/mL) or Atsttrin (500 ng/ml). As indicated in Fig. 1c. Western blotting analysis indicated that tartrate-resistant acid phosphatase (TRAP) and cathepsin K (CTSK) expression was increased in the presence of TNF-α, while Atsttrin dramatically reduced TNF-α and RANKL-induced osteoclastogenesis. Importantly, as illustrated in Fig. 1d, e, qualification of the bands indicated that Atsttrin significantly reduced TNF-α- and RANKL-mediated TRAP and CTSK expression. To further determine the role of Atsttrin in osteoclastogenesis, we cultured Raw264.7 cells, as previously described, for 24 h and extracted mRNA, followed by real-time PCR assay. As shown in Fig. 1f–h. TNF-α significantly upregulated the transcriptional levels of TRAP, CTSK, and Calcitonin Receptor. Meanwhile, Atsttrin effectively reduced the expression of TNF-α-mediated osteoclastogenesis gene markers.

### Atsttrin inhibited TNF-α-mediated inflammatory catabolism.
TNF-α is believed to have an inflammatory catabolic effect in the pathogenesis of osteoporosis[17,26]. To further determine the antagonistic effect of Atsttrin on TNF-α-mediated inflammatory responses, we cultured RAW264.7 cells and BMDMs with or without TNF-α (10 ng/ml) in the absence or presence of Atsttrin (500 ng/ml) to examine the expression of downstream inflammatory factors (IL-1β, IL-6, COX-2, and iNOS). As shown in Fig. 2a–i, TNF-α dramatically increased the expression of inflammatory factors (IL-1β, IL-6, COX-2, and iNOS), while Atsttrin reduced the mRNA and protein expression of these molecules. To further investigate the molecular mechanism involved, we cultured RAW264.7 cells with or without TNF-α (10 ng/ml) in the absence or presence of Atsttrin (500 ng/ml) for various durations. As demonstrated in Fig. 2j–n, TNF-α effectively phosphorylated p38, JNK, IκBα, and p65 in a time-dependent manner. However, Atsttrin significantly inhibited TNF-α-induced phosphorylation of p38, JNK, and p65 and the degradation of IκBα. To further investigate whether Atsttrin inhibited TNF-α-induced p65 translocation, RAW264.7 cells were cultured with TNF-α (10 ng/mL) in the absence or presence of Atsttrin (500 ng/ml) for various time points. Nuclear extraction (NE) and cytoplasmic extraction (CE) were examined by Western blotting. As shown in Fig. 2o–q, the total expression of P65 in the cytoplasm was decreased, while that of P65 in the nucleus was increased with time in the presence of TNF-α. Collectively, Atsttrin effectively inhibited TNF-α-mediated inflammatory catabolism through the TNF-α/NF-κB signaling pathway.

### Atsttrin inhibited TNF-α-induced osteoclastogenesis via TNFR1.
TNF-α exerts its effects through TNF-α receptors, including TNFR1 and TNFR2[27,28]. TNFR1 is reported to promote inflammatory effects, and TNFR2 is reported to exert protective effects[21]. We next sought to determine whether Atsttrin mediated this effect through TNFR1 or TNFR2. Thus, we suppressed the expression of TNFR1 using its specific siRNAs in RAW264.7 cells and BMDMs, and as indicated in Fig. 3a, we found that TNFR1 RNAi dramatically inhibited TNFR1 expression. In addition, we found that after knocking down TNFR1 in

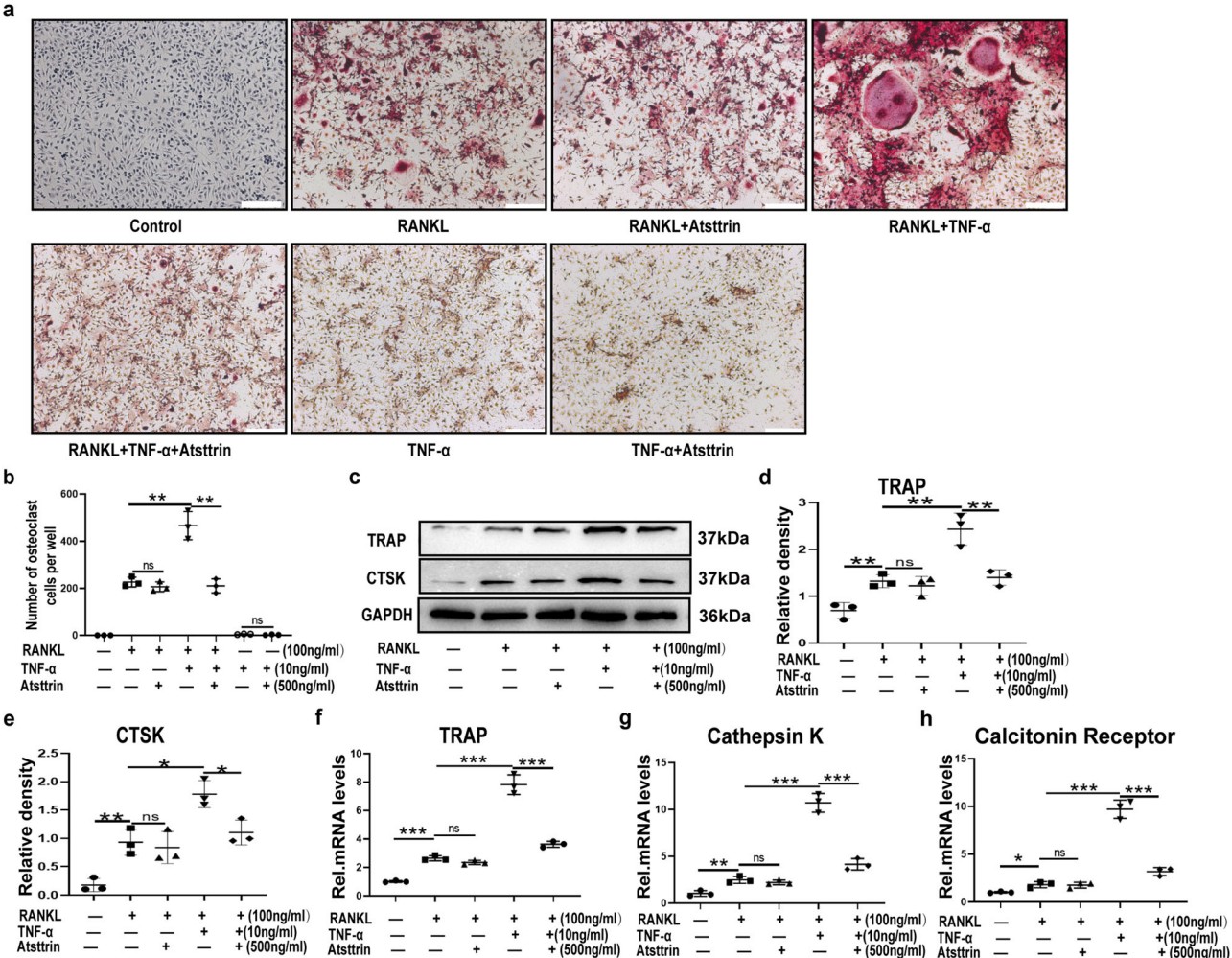

**Fig. 1 Atsttrin inhibited TNF-α-induced osteoclastogenesis. a** BMDMs were treated with M-CSF (10 ng/mL) for 6 days and then cultured with RANKL (100 ng/mL) and without or with TNF-α (10 ng/ml) in the absence or presence of Atsttrin (500 ng/ml) for 7 days and TRAP staining was performed. Scale bar, 200 μm. Each experiment was performed 3 times independently. **b** TRAP-positive multinuclear cells with more than three nuclei were considered mature osteoclasts (OCs). Osteoclast cell numbers were quantified. **c** BMDMs were treated with RANKL (100 ng/mL), TNF-α (10 ng/ml), and Atsttrin (500 ng/ml) for 48 h. Protein expression levels of TRAP and CTSK were measured by Western blot ($n = 3$). The bands in the figure are not all derived from the same membrane. **d, e** Western blot gray value analysis of TRAP in BMDMs. **f–h** RAW264.7 cells were treated with RANKL (100 ng/mL), TNF-α (10 ng/ml), and Atsttrin (500 ng/ml) for 24 h. The mRNA expression levels of TRAP, Cathepsin K, and the Calcitonin receptor were measured by real-time PCR. Each experiment was performed 3 times independently. Significant differences are indicated as follows: $^{ns}P > 0.05$, $*P < 0.05$, $**P < 0.01$ and $***P < 0.001$.

BMDMs, TNF-α-mediated TRAP and CTSK expression were dramatically reduced (Fig. 3b–d). Importantly, Atsttrin-mediated inhibition of TRAP expression was also almost abolished (Fig. 3b–d). Moreover, as revealed in Fig. 3e, f, TRAP staining showed a significant reduction in osteoclast formation after TNFR1 knockdown. A real-time PCR assay was performed to investigate the involved mechanisms. As revealed in Fig. 3g–i, after the blockage of TNFR1, Atsttrin-mediated inhibition of TNF-α was almost abolished. These findings indicated that Atsttrin mediated anti-osteoclastogenesis primarily through TNFR1 signaling.

**Atsttrin rescues TNF-α-mediated inhibition of osteoblastogenesis through TNFR1.** TNF-α is believed to significantly inhibit osteoblastogenesis[29]. Therefore, to investigate whether Atsttrin rescues TNF-α-mediated inhibition of osteoblastogenesis, BMMSCs were cultured in the presence of osteoblastogenesis medium with or without TNF-α (10 ng/mL) in the absence or presence of Atsttrin (500 ng/ml) for 7 days, followed by ALP staining. As revealed in Fig. 4a, b, TNF-α effectively inhibited osteoblast formation, while Atsttrin significantly rescued this pathogenesis. In addition, to determine the protein level change, total protein was extracted and followed by Western blotting analysis. As shown in Fig. 4c, d, RUNX-2 expression was significantly reduced in the presence of TNF-α, while Atsttrin effectively attenuated this process. To investigate the transcriptional changes involved, we next cultured MC3T3-E1 cells with 10 ng/mL TNF-α with or without Atsttrin for 8 hours. Total mRNA was extracted, and a real-time PCR assay was performed. As illustrated in Fig. 4e–g, Atsttrin alleviated TNF-α-mediated down-regulation of the bone formation markers, including ALP, RUNX2, and Col-1. To further investigate the molecular mechanism involved, MC3T3-E1 cells were cultured in the absence or presence of 10 ng/ml TNF-α with or without 500 ng/ml Atsttrin for different durations. Western blot analysis indicated that Atsttrin downregulated TNFα-activated signaling pathways (Fig. 4h–k). Collectively, these results indicated that Atsttrin rescued TNF-α-mediated inhibition of osteoblast formation.

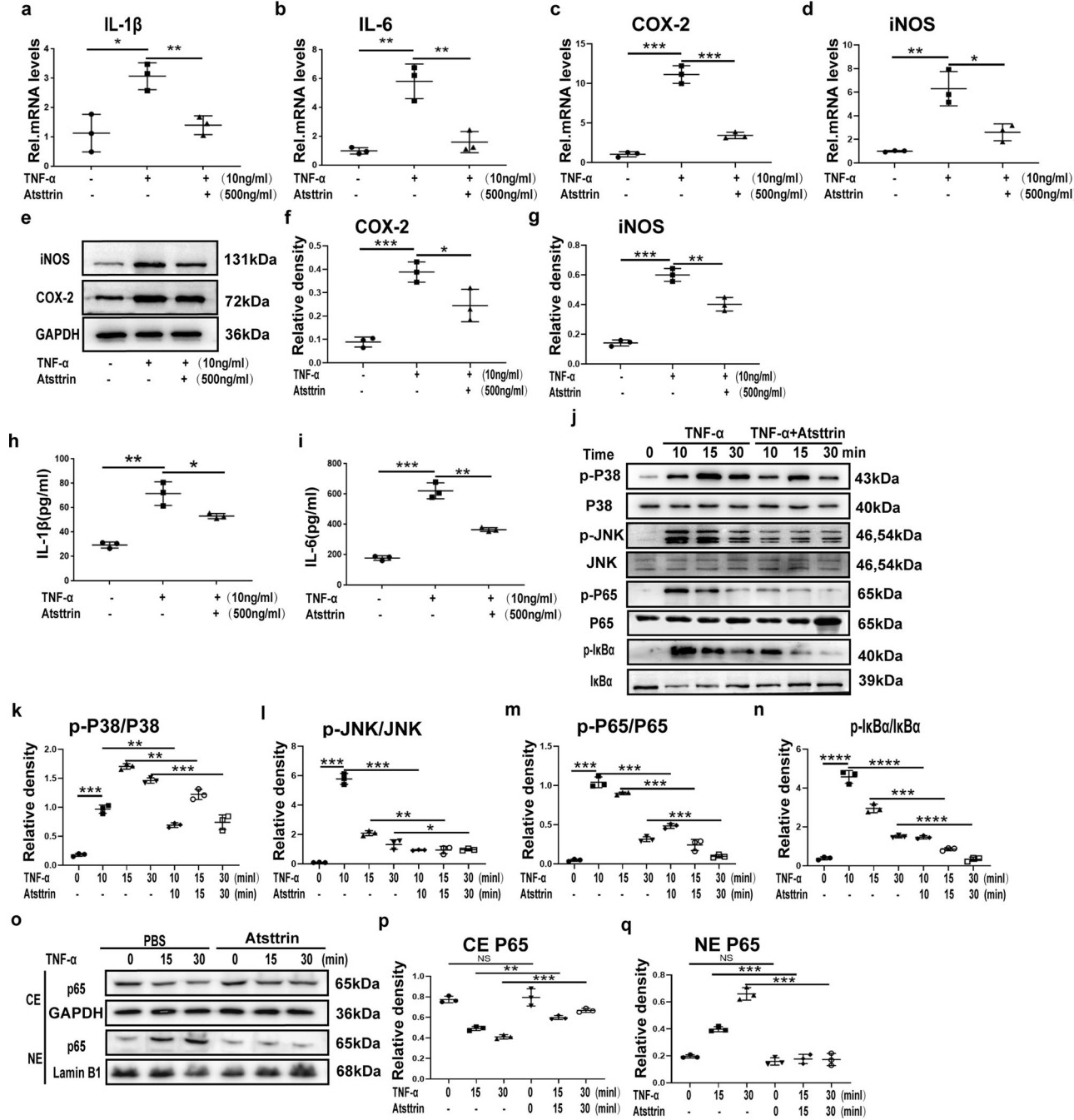

**Fig. 2 Atsttrin inhibited TNF-α-mediated inflammation. a–d** RAW264.7 cells were treated with TNF-α (10 ng/ml) and Atsttrin (500 ng/ml) for 8 h. mRNA expression of inflammatory molecules (IL-1β, IL-6, COX-2, and iNOS) was tested by real-time PCR (n = 3). **e–g** RAW264.7 cells treated with 10 ng/ml TNF-α without or with 200 ng/ml Atsttrin. The expression of COX-2 and iNOS was tested by Western blot assay. Quantitative analyses of the normalized protein levels (n = 3). **h, i** BMDMs were treated with M-CSF (10 ng/mL) for 6 days and then cultured with or without TNF-α (10 ng/mL) in the absence or presence of Atsttrin (500 ng/ml) for 48 h. The levels of IL-1β and IL-6 in the medium were detected by ELISA (n = 3). **j–n** RAW264.7 cells were treated with TNF-α (10 ng/mL) in the absence or presence of Atsttrin (500 ng/ml) for various durations. Protein levels of p-p38, t-p38, p-JNK, t-JNK, p-p65, t-p65, p-IκBα, and IκBα were detected by Western blot with corresponding antibodies (n = 3). The bands in the figure are not all derived from the same membrane. **o–q** RAW264.7 cells were treated with TNF-α (10 ng/mL) in the absence or presence of Atsttrin (500 ng/ml) for various time points. Nuclear translocation of NF-κB p65 in RAW264.7 cells as determined by Western blotting with an anti-p65 antibody (n = 3). Each experiment was performed 3 times independently. Significant differences are indicated as follows: $^{ns}P > 0.05$, $*P < 0.05$, $**P < 0.01$, $***P < 0.001$ and $****P < 0.0001$.

Given the importance of TNFR signaling as well as the nature of Atsttrin as the binding protein of TNFRs, we next determined whether Atsttrin exhibited this protective effect through the TNFR pathway. To address this issue, we suppressed the expression of TNFR1 using specific siRNAs in MC3T3-E1 cells and BMMSCs (Fig. 5a). As illustrated in Fig. 5b, c, TNF-α dramatically reduced the expression of RUNX2; however, the expression of RUNX2 was restored after blocking TNFR1. Consequently, as shown in Fig. 5d, e, cell alkaline phosphatase (ALP) staining of osteoblasts indicated that osteoblast formation

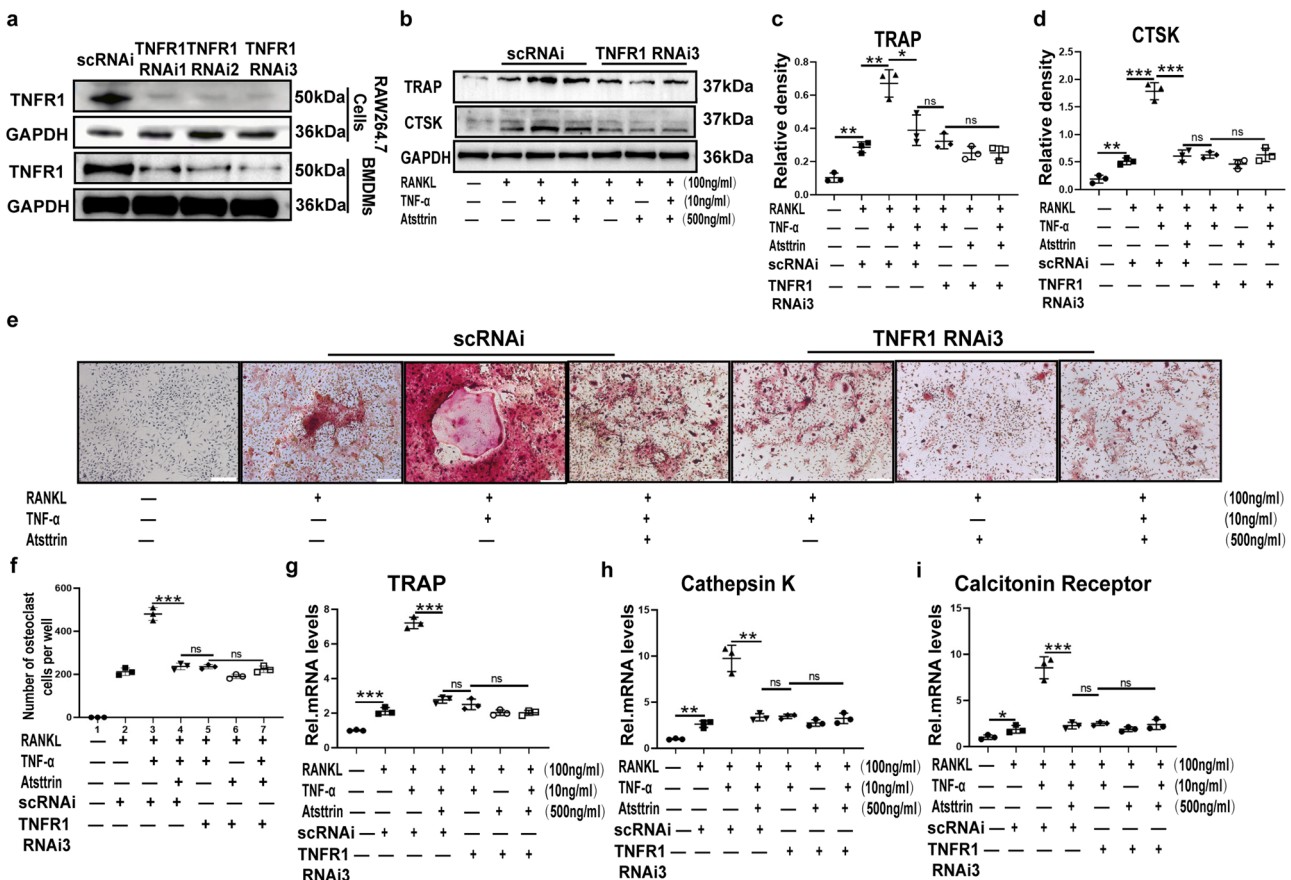

**Fig. 3 Atsttrin inhibited TNF-α-induced osteoclastogenesis via TNFR1. a** Western blot analysis to examine the knockdown efficacy of siRNA against TNFR1 in RAW264.7 cells and BMDMs. **b** BMDMs transfected with scrambled control siRNA (scRNAi) or TNFR1 RNAi were treated with RANKL (100 ng/ml), TNF-α (10 ng/mL), and Atsttrin (500 ng/ml) for 48 h. The protein levels of TRAP and CTSK were measured by Western blotting ($n = 3$). The bands in the figure are not all derived from the same membrane. **c, d** Western blot gray value analysis of TNFR1 and TRAP in BMDMs. **e, f** BMDMs transfected with scrambled control siRNA (scRNAi) or TNFR1 RNAi were cultured with RANKL (100 ng/ml), TNF-α (10 ng/ml), and Atsttrin (500 ng/ml) for 7 days, and TRAP staining was performed. Scale bar, 200 μm. The TRAP-positive multinuclear cells were counted. Each experiment was performed three times independently. **g–i** RAW264.7 cells transfected with scrambled control siRNA (scRNAi) or TNFR1 RNAi were treated with RANKL (100 ng/ml), TNF-α (10 ng/ml), and Atsttrin (500 ng/ml) for 8 h. The mRNA levels of TRAP, Cathepsin K, and Calcitonin Receptor were measured by real-time PCR ($n = 3$). Significant differences are indicated as follows: $^{ns}P > 0.05$, $*P < 0.05$, $**P < 0.01$ and $***P < 0.001$.

was reduced in the presence of TNF-α. However, the knockdown of TNFR1 remarkably reversed TNF-α-inhibited osteoblast formation. Importantly, additional use of Atsttrin even slightly enhanced osteoblastogenesis after TNFR1 blockage. To further examine gene marker expression, we cultured MC3T3-E1 cells for 8 h and collected total mRNA. As demonstrated in Fig. 5f–h, the transcriptional levels of ALP, RUNX2, and Col-1 were downregulated in the presence of TNF-α. However, this phenomenon was reversed when TNFR1 was knocked down. Thus, these findings indicated that Atsttrin exhibited a protective effect by inhibiting TNF-α-mediated inhibition of osteoblastogenesis primarily through the TNFR1 pathway.

**Atsttrin enhances osteoblastogenesis through TNFR2.** Given the importance of osteoblasts in osteoporosis, we next determined whether Atsttrin has an anabolic effect on osteoblastogenesis in basic medium. The present study found that Atsttrin alone weakly activated Erk1/2 or Akt signaling (Supplementary Fig. 1e). Consequently, the bone formation protein and transcriptional expression of RUNX2, ALP, and Col-1 were not altered in the presence of Atsttrin (Supplementary Fig. 1f–i). In addition, to examine whether Atsttrin and osteoblastogenesis medium have a

synergistic effect on osteoblast formation, MC3T3-E1 cells were cultured in the presence of osteoblastogenesis medium, with or without Atsttrin. To address this issue, 200 ng/ml and 500 ng/ml Atsttrin were used to stimulate cells. As illustrated in Supplementary Fig. 1d, 200 ng/ml Atsttrin weakly activated Erk1/2 or Akt signaling. In contrast, 500 ng/ml Atsttrin dramatically enhanced Erk1/2 and Akt signaling in the osteoblastogenesis medium. In this case, the present study used 500 ng/ml Atsttrin in the in vitro study to make it consistent. As shown in Fig. 6a, b, Atsttrin significantly enhanced osteoblastogenesis in the presence of osteoblastogenesis medium. In addition, alkaline phosphatase (ALP) staining indicated that more osteoblast formation was observed in the presence of Atsttrin (Fig. 6c, d). Real-time PCR was carried out, which showed that Atsttrin treatment enhanced the transcriptional expression of bone formation markers, including ALP, RUNX2, and Col-1 (Fig. 6e, g). Importantly, ALP activity demonstrated that Atsttrin enhanced not only osteoblastogenesis but also osteoblast activity (Fig. 6h).

To determine whether the Atsttrin-mediated enhancement in osteoblastogenesis is dependent upon TNFR1, TNFR2, or both receptors, we suppressed the expression of TNFR2 using specific siRNAs in MC3T3-E1 cells and BMMSCs (Fig. 6i). MC3T3-E1 cells were collected from different experimental groups, Western

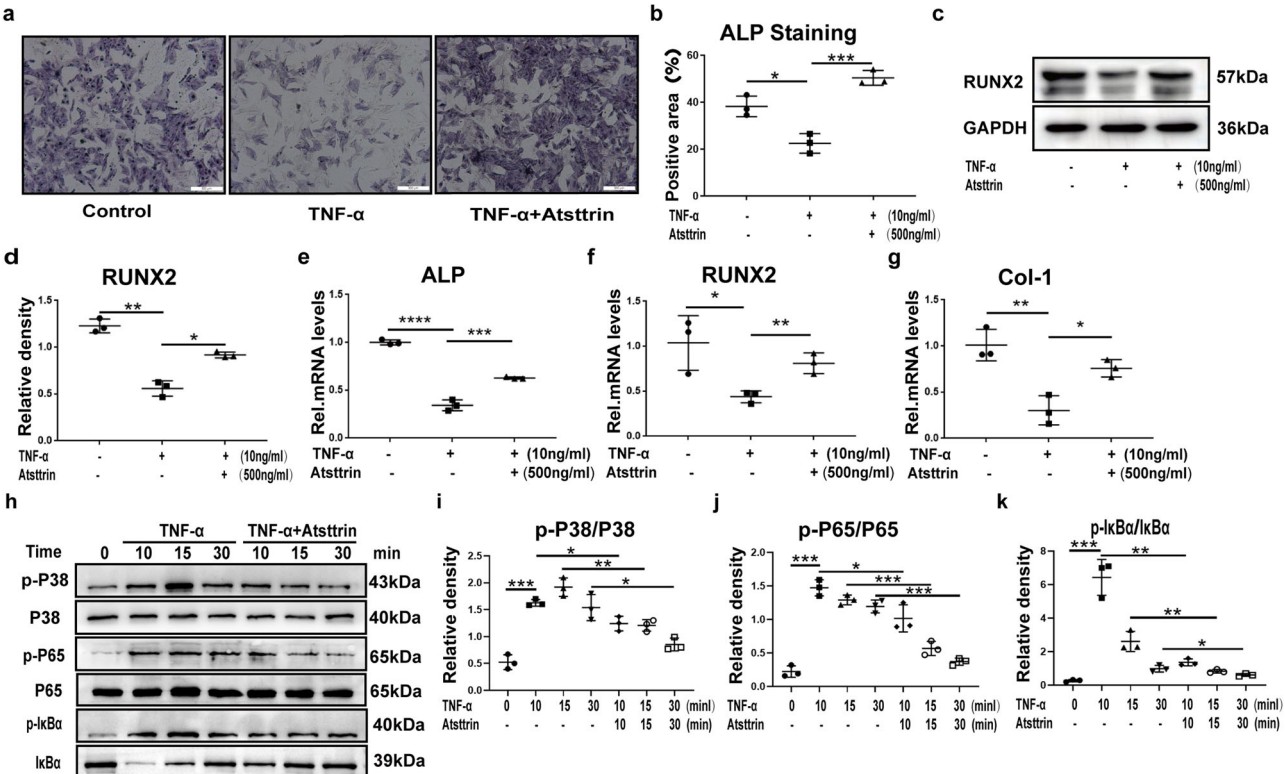

**Fig. 4 Atsttrin alleviated TNF-α-mediated inhibition of osteoblastogenesis. a, b** BMMSCs were cultured without or with TNF-α (10 ng/ml) in the absence or presence of Atsttrin (500 ng/ml) for 7 days, and ALP staining was performed. Scale bar, 200 μm. Each experiment was performed 3 times independently. Quantification of the percentage of the positive area was based on ALP staining. **c, d** MC3T3-E1 cells were treated with TNF-α (10 ng/ml) and Atsttrin (500 ng/ml) for 48 h in an osteogenesis-induction culture medium. The protein expression of RUNX2 was tested by Western blotting ($n = 3$). Quantification of the band density for RUNX2 based on the Western blotting assay. **e–g** MC3T3-E1 cells were treated with TNF-α (10 ng/mL) and Atsttrin (500 ng/ml) for 8 h in an osteogenesis-induced culture medium, as indicated. The mRNA levels of ALP, RUNX2, and Col-1 were measured by real-time PCR ($n = 3$). **h–k** Western blotting and quantitative analyses of the phosphorylation of p38, p65, and IκBα ($n = 3$). The bands in the figure are not all derived from the same membrane. Significant differences are indicated as follows: *$P < 0.05$, **$P < 0.01$, ***$P < 0.001$, and ****$P < 0.0001$.

blot to detect RUNX2(Fig. 6j, k), real-time PCR to detect ALP, RUNX2 and Col-1 (Fig. 6n–p), In addition, alkaline phosphatase (ALP) staining was used to detect the expression of ALP (Fig. 6l–m). We found that suppression of TNFR2 led to almost abolishment of the Atsttrin-enhanced osteoblastogenesis.

**Atsttrin-mediated enhancement of osteoblastogenesis primarily depends on TNFR2-Akt-Erk1/2 signaling.** Atsttrin is known to selectively bind to TNFRs, and the Atsttrin-mediated beneficial effect on osteoblastogenesis depends on TNFR2. We next sought to explore the molecular mechanisms. A previous study indicated that Atsttrin activated Akt and Erk1/2 signaling in chondrocytes[20,21]. Interestingly, the present study demonstrated that Atsttrin strongly enhanced the activation of Akt signaling and Erk1/2 signaling in the osteoblastogenesis medium (Fig. 7a–c and Supplementary Fig. 1d).

To further determine whether Atsttrin-mediated enhancement relies on Akt and/or Erk1/2 signaling, MC3T3-E1 cells were cultured with osteoblastogenesis medium in the absence or presence of Atsttrin (500 ng/ml) with or without PI3K/Akt-signaling inhibition using 1 μM wortmannin or Erk1/2 signaling inhibition using 1 μM U0126 for 8 h, followed by real-time PCR assay. As indicated in Fig. 7d–i, the positive effect of Atsttrin was partially lost after blocking Akt signaling alone using wortmannin or blocking the Erk1/2 signaling pathway alone using wortmannin. Furthermore, by applying both Akt and Erk1/2 signaling inhibitors together, as shown in Fig. 7j–l, we found that the Atsttrin-mediated positive effect on osteoblastogenesis was

almost abolished. Collectively, Atsttrin-mediated enhancement of osteoblastogenesis depended on the TNFR2/Akt and Erk1/2 pathways.

**Atsttrin attenuates bone loss in ovariectomy-induced mice.** The above in vitro experiments confirmed the role of Atsttrin in osteoclastogenesis and osteoblastogenesis. To investigate whether Atsttrin could prevent accelerated bone loss in ovariectomy-induced mice. Mice received an intraperitoneal injection of Atsttrin or PBS weekly for 8 weeks after OVX surgery. As shown in Fig. 8a–d, the quantitative analysis indicated that Atsttrin effectively protected against bone loss after OVX surgery. Specifically, Atrsttrin manifests as an increase in BMD, Tb.N, and BV/TV. In addition, as shown in Fig. 8e, f, the application of recombinant Atsttrin also significantly reduced the levels of bone catabolism indicators (TRAP and CTSK) in peripheral blood serum as assayed by ELISA. Together, these studies indicate that Atsttrin is a therapeutic candidate for reducing bone loss in ovariectomy-induced mice.

## Discussion

Osteoporosis is a common degenerative disease in clinical practice. With the aging of the global population, the prevalence of osteoporosis is increasing every year, resulting in substantial bone-related morbidities and increased mortality and healthcare costs[30]. However, the underlying mechanisms involved in osteoporosis remain unclear. Under physiological conditions, bone metabolism is maintained through a delicate balance

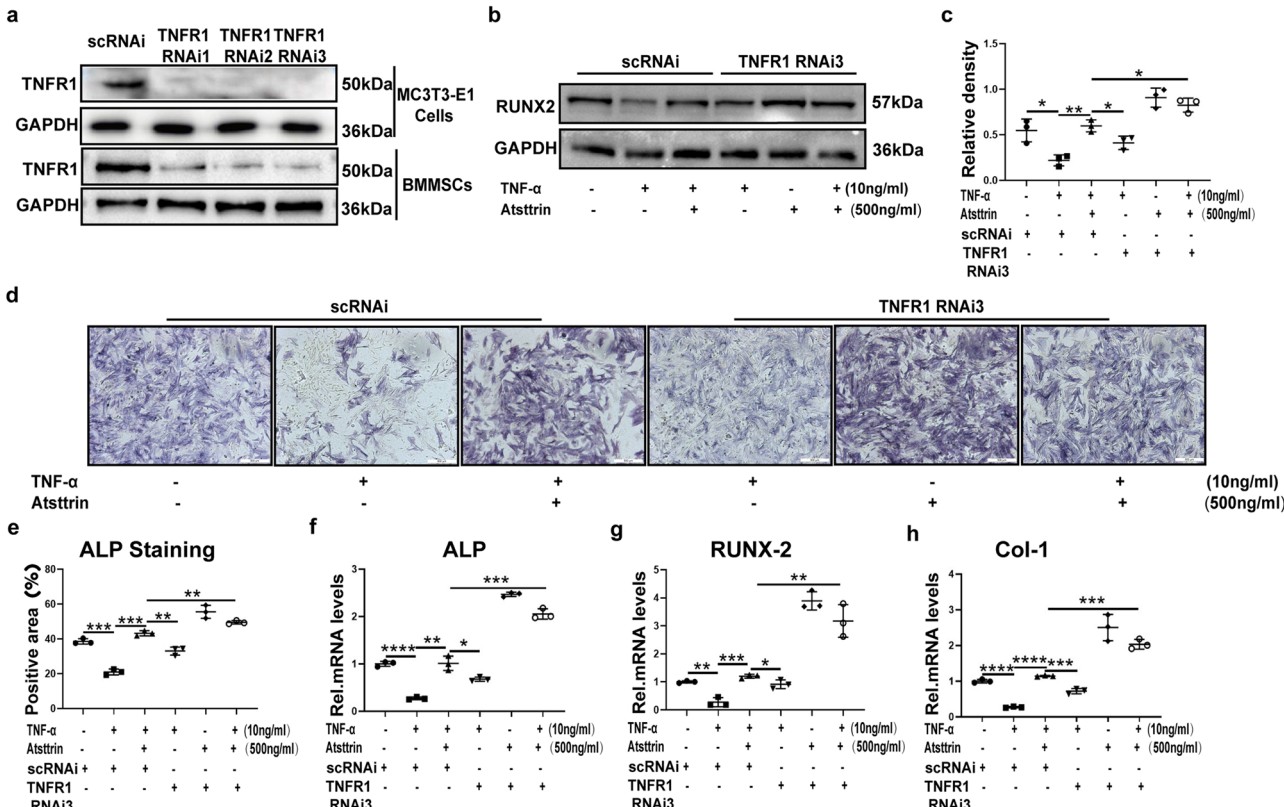

**Fig. 5 Atsttrin rescues TNF-α-mediated inhibition of osteoblastogenesis through TNFR1. a** Immunoblotting analysis to examine the knockdown efficacy of siRNA against TNFR1. **b**, **c** BMMSCs transfected with scRNAi or TNFR1 RNAi were treated with TNF-α (10 ng/mL) and Atsttrin (500 ng/ml) for 48 h. The protein levels of RUNX2 were measured by Western blotting. **d**, **e** BMMSCs transfected with scRNAi or TNFR1 RNAi were treated with TNF-α (10 ng/mL) and Atsttrin (500 ng/ml) for 7 days, and ALP staining was performed. Scale bar, 200 μm. Each experiment was performed 3 times independently. Quantification of the percentage of the positive area was based on ALP staining. **f–h** MC3T3-E1 cells transfected with scRNAi or TNFR1 RNAi were treated with TNF-α (10 ng/mL) and Atsttrin (500 ng/ml) for 8 h. The transcriptional levels of ALP, RUNX2, and Col-1 were measured by real-time PCR ($n = 3$). Significant differences are indicated as follows: $*P < 0.05$, $**P < 0.01$, $***P < 0.001$ and $****P < 0.0001$.

between bone formation and bone resorption. Nevertheless, this balance is disrupted in osteoporosis[31]. Previous reports have shown that inflammation plays an important role in the development of osteoporosis[32]. TNF-α is located at the top of the inflammatory cascade and plays an important role in the occurrence of osteoporosis[16]. TNF-α can lead to decreased activity of osteoblasts and increased number and activity of osteoclasts, resulting in bone loss[33].

In osteoarthritis, inflammatory bowel disease, and psoriasis, PGRN can effectively inhibit the inflammatory signaling pathway mediated by TNF-α and reduce the expression of downstream inflammatory factors, thus inhibiting the inflammatory response mediated by TNF-α[18]. In TNF-α transgenic mice, TNF-α overexpression resulted in significant bone loss, which was significantly reduced by the application of exogenous PGRN[34]. Moreover, PGRN can mediate bone formation induced by BMP-2[34,35]. Further studies have shown that PGRN can promote fracture healing under physiological conditions[11,19]. It was hypothesized that Atsttrin, as an engineered protein derived from PGRN, would regulate bone metabolism balance through a mechanism highly similar to that of PGRN. Previous studies have shown that Atsttrin has an effect on osteoclastogenesis at a concentration of 100 ng/ml[36]. To address the issue of Atsttrin's concentration effect on osteoclast and osteoblast formation, we established an in vitro cell model with various concentrations. As indicated in Supplementary Fig. 1a–c, with a concentration of 42.35 ng/ml (2.5 nM), and 50 ng/ml (2.95 nM), Atsttrin exerted a sufficient effect on anti-TNF-α. In addition, Atsttrin showed an

anti-catabolic effect in a dose-dependent manner with concentrations of 100 ng/ml (5.9 nM), 200 ng/ml (11.9 nM), and 500 ng/ml (29.5 nM), respectively. Especially with a concentration of 1000 ng/ml (59 nM), Atsttrin almost abolished the effect of TNF-α. The concentration of 2.95 nM was similar to the previous publication[10]. To determine the concentration of Atsttrin for activating osteoblast formation, the present study took advantage of 200 ng/ml (11.9 nM) and 500 ng/ml (29.5 nM) Atsttrin. As illustrated in Supplementary Fig. 1d, 200 ng/ml (11.9 nM) Atsttrin weakly activates Erk1/2 or Akt pathway, while 500 ng/ml (29.5 nM) Atsttrin could strongly activate Erk1/2 and Akt pathway. To make the concentration consistent in the whole manuscript, the present study took a concentration of 500 ng/ml (29.5 nM) to investigate the role of Atsttrin in osteoblast and osteoclast formation. The safety of Atsttrin was our foremost consideration, and we designed relevant tests in the current study to ensure the safety of working concentrations. As demonstrated in Supplementary Fig. 1j, k, 500 ng/ml (29.5 nM), Atsttrin could not affect cell viability. Taking all concerns into account, the present study used 500 ng/ml Atsttrin in the in vitro study. Next, BMDMs were treated with receptor activators of nuclear factor kappa-B ligand (RANKL) (100 ng/mL) for 7 days to induce osteoclast differentiation[37,38]. We found that TNF-α can enhance the activity of osteoclasts and promote the proliferation of osteoclasts, while Atsttrin attenuated this progress. Furthermore, we established a TNF-α-induced inflammatory response model in RAW264.7 cells to explore the role of Atsttrin in inhibiting osteoclastogenesis and to investigate the mechanism underlying

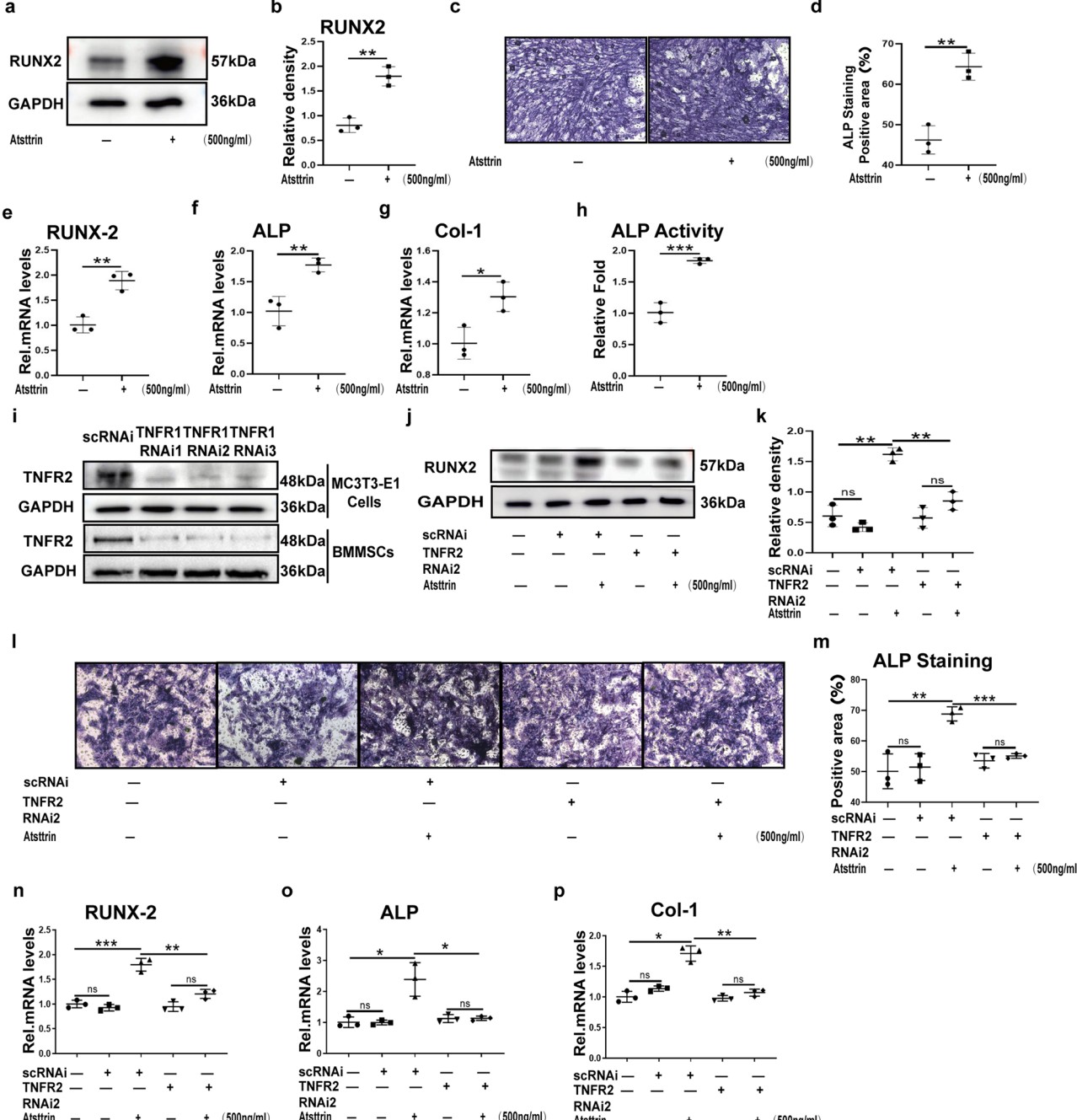

**Fig. 6 Atsttrin enhances osteoblastogenesis through TNFR2. a, b** MC3T3-E1 cells were treated with Atsttrin (500 ng/ml) for 48 h. The protein level of RUNX2 was assessed by Western blotting ($n = 3$). Quantification of the band density for RUNX2 based on the Western blotting assay. **c, d** BMMSCs were treated with Atsttrin (500 ng/ml) for 7 days, and ALP staining was performed. Scale bar, 200 μm. Each experiment was performed three times independently. Quantification of the percentage of the positive area was based on ALP staining. **e–g** MC3T3-E1 cells were cultured with Atsttrin (500 ng/ml) for 8 h. The mRNA levels of ALP, RUNX2, and Col-1 were detected by real-time PCR ($n = 3$). **h** MC3T3-E1 cells were treated with Atsttrin (500 ng/ml) for 7 days. The relative fold ALP activity was detected by an ALP assay kit ($n = 3$). **i** Knockout efficiency of TNFR2 using siRNA in MC3T3-E1 cells and BMMSCs, as assayed by immunoblotting analysis. **j, k** MC3T3-E1 cells transfected with scRNAi or TNFR2 RNAi were treated with Atsttrin (500 ng/ml) for 48 h. The protein was examined by Western blotting with an anti-RUNX2 antibody ($n = 3$). Quantification of the band density for RUNX2 based on the Western blotting assay. **l, m** BMMSCs transfected with scRNAi or TNFR2 RNAi were treated with Atsttrin (500 ng/ml) for 7 days, and ALP staining was performed. Scale bar, 200 μm. Each experiment was performed three times independently. Quantification of the percentage of the positive area was based on ALP staining. **n–p** MC3T3-E1 cells transfected with scRNAi or TNFR2 RNAi were treated with Atsttrin (500 ng/ml) for 8 h. The mRNA levels of ALP, RUNX2, and Col-1 were measured by real-time PCR ($n = 3$). Significant differences are indicated as follows: ${}^{ns}P > 0.05$, $*P < 0.05$, $**P < 0.01$ and $***P < 0.001$.

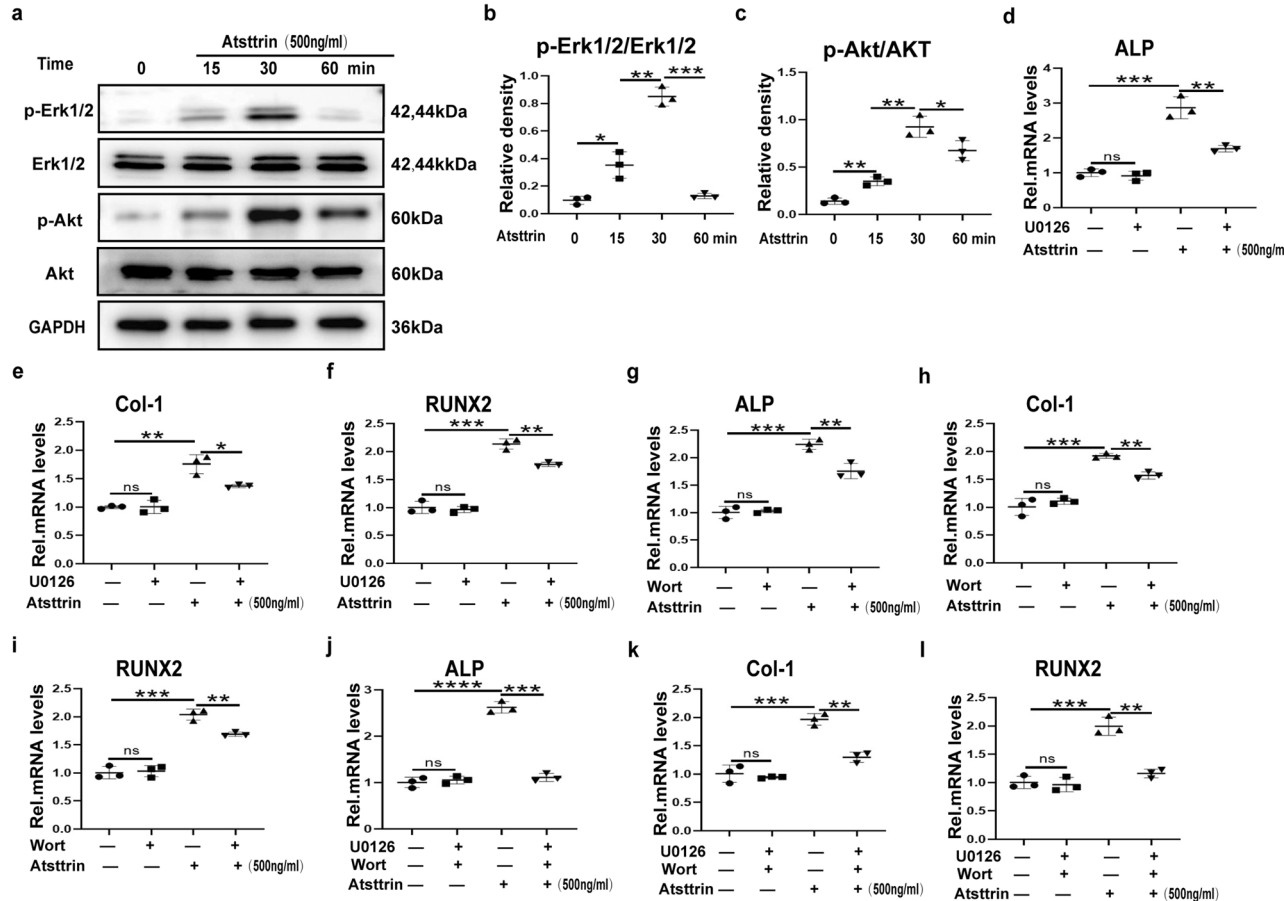

**Fig. 7 Atsttrin-mediated enhancement of osteoblastogenesis primarily depends on TNFR2-Akt-Erk1/2 signaling. a–c** MC3T3-E1 cells were treated with Atsttrin (1000 ng/ml) and collected at various time points, followed by Western blot analysis using ImageJ ($n = 3$). The bands in the figure are not all derived from the same membrane. **d–l** MC3T3-E1 cells were cultured without or with Atsttrin in the absence or presence of 1 μM Erk1/2 signaling blocker U0126 and/or 1 μM Akt signaling blocker wortmannin. The transcriptional levels of ALP, RUNX2, and Col-1 in MC3T3-E1 cells were tested by real-time PCR ($n = 3$). Significant differences are indicated as follows: $^{ns}P > 0.05$, $^*P < 0.05$, $^{**}P < 0.01$, $^{***}P < 0.001$ and $^{****}P < 0.0001$.

this effect. We found that Atsttrin effectively prevented the TNF-α-mediated inflammatory catabolism signaling pathway, as evidenced by the significantly suppressed expression levels of IL-1β, IL-6, COX-2, and iNOS and reduced serum levels of IL-1β and IL-6 fragments. Moreover, TNF-α is involved in osteoclastogenesis mainly through activation of the NF-κB, p38, and JNK signaling pathways. The levels of p-P38, p-JNK, and p-P65 were significantly reduced by Atsttrin treatment. In addition, the NF-κB signaling pathway plays a critical role in TNF-α function. Nuclear translocation of p65 can be inhibited after treatment with Atsttrin.

TNF-α plays an important role in the pathogenesis of osteoporosis and is a major inflammatory factor. There are two binding sites of TNF-α on the cell surface, TNF receptor-1 and TNF receptor-2. The interaction between TNF-α and TNFR1 mediates inflammatory signaling[39,40]. TNF-α induces the expression of molecules associated with inflammation in osteoclast precursors, including IL-1β, IL-6, COX-2, and iNOS, through binding to its receptor TNFR1. We also assessed the expression of biomarkers in TNF-α-induced osteoclastogenesis. In addition, TNF-α inhibits osteoblast precursor differentiation through TNFR1. The additional use of Atsttrin to block TNFR1 prominently repressed this process. Previous reports have demonstrated that TNFR2 mediates protective signaling[41]. We found that Atsttrin can enhance osteoblastogenesis through binding to TNFR2. Therefore, during the progression of

osteoporosis, we found that Atsttrin exhibited its effect through both inhibiting TNFα/TNFR1-mediated inflammation and activating anabolic TNFR2 pathways. We found that Erk1/2 and Akt signaling are involved in osteoblastogenesis. In summary, in the presence of osteoblastogenesis medium, Atsttrin strongly activates Akt/Erk1/2 signaling. However, in normal media, Atsttrin can weakly activate the Akt/Erk1/2 signaling pathways. Furthermore, Atsttrin completely lost its effect when using specific inhibitors of Akt and Erk1/2 signaling. Consequently, we found that the Atsttrin-mediated anabolic effect in osteoblastogenesis depends on TNFR2-Akt/Erk1/2 signaling. Our study provides new evidence to improve the understanding of how TNFR functions in the regulation of bone homeostasis and the specific mechanisms involved in osteoporosis. This study identified TNFR as a crucial target for osteoporosis, thus advancing the current understanding of its mechanism of action in the regulation of bone homeostasis.

The ovariectomy-induced mouse model is an established research method for studying the pathogenesis of osteoporosis in vivo[42,43]. Notably, we found that intraperitoneal injection of recombinant Atsttrin could dramatically prevent bone loss in the VOX model. In the current study, we confirmed the role of Atsttrin in osteoclastogenesis and osteoblastogenesis. Although Atsttrin is derived from PGRN, whether PGRN can prevent osteoporosis has not been reported and needs further investigation.

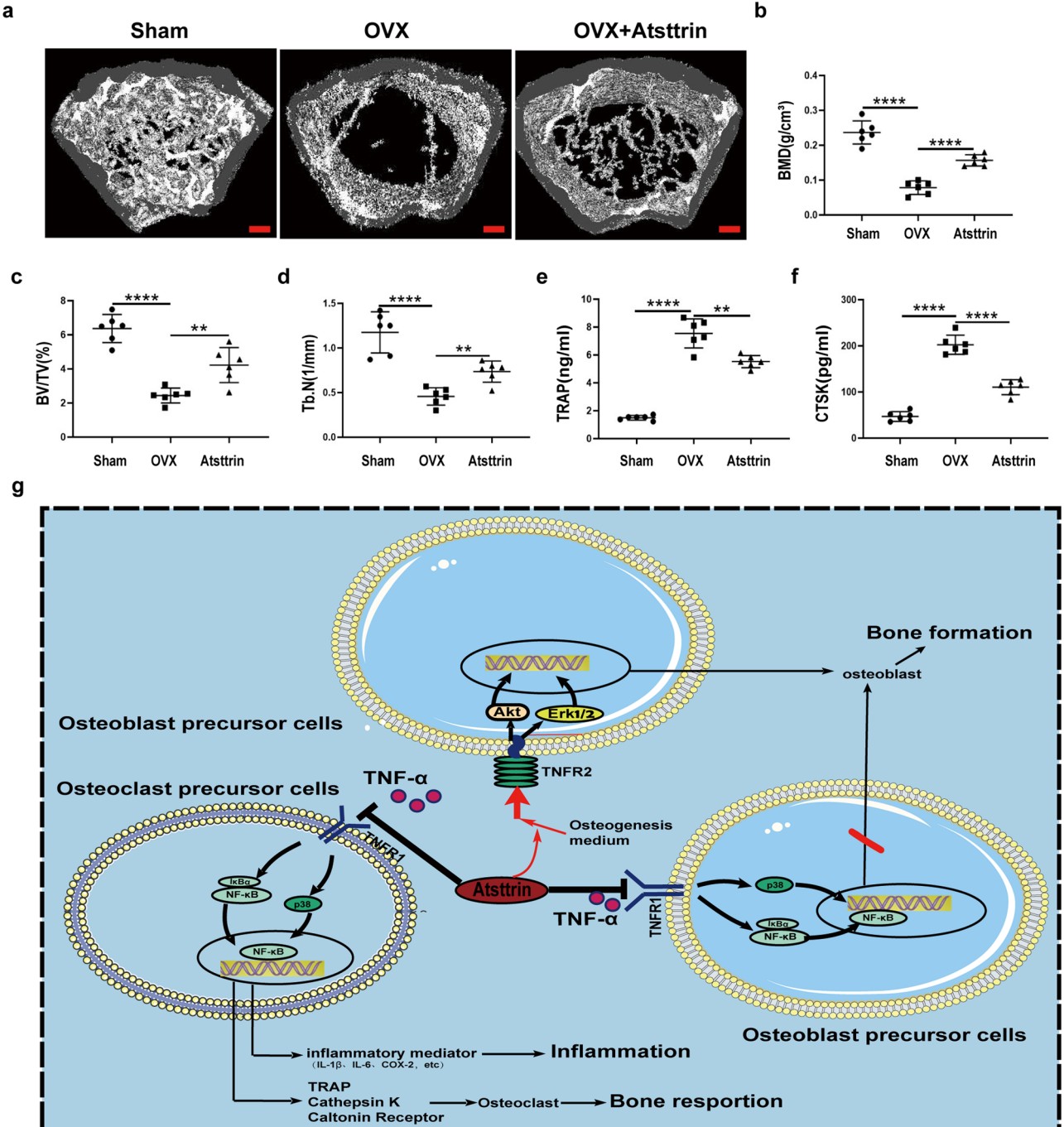

**Fig. 8 Atsttrin attenuates bone loss in the OVX model. a** Representative micro-CT images of bone trabeculae of distal femurs. Scale bars, 500 μm. **b–d** Quantitative analysis of the BMD, Tb.N, and BV/TV of the distal femurs ($n = 6$). **e, f** The levels of TRAP and CTSK in mouse serum were detected by ELISA. **g** A proposed model for explaining the effect of Atsttrin on regulating osteogenic-osteoclast balance through the TNFR signaling pathway in the treatment of osteoporosis. Significant differences are indicated as follows: $^{**}P < 0.01$ and $^{****}P < 0.0001$.

Overall, Atsttrin enhances osteoblastogenesis and inhibits osteoclastogenesis under inflammatory conditions through at least 3 pathways: a) Atsttrin directly binds to TNFR1 and inhibits TNF-α-induced osteoclastogenesis; b) Atsttrin competitively binds to TNFR1 and rescues TNF-α-mediated inhibition of osteoblastogenesis; and c) Atsttrin synthesized with osteoblastogenesis medium, enhanced osteoblast formation and activity via TNFR2-Akt/Erk1/2 signaling (Fig. 8g). There is an interaction between bone formation and bone resorption that limits the pharmacologic treatment of osteoporosis to some extent[44]. Antiresorptive drugs targeting osteoclasts induce a decrease in osteoblast activity, and the ability of the bone formation-promoting drug teriparatide to promote bone formation is partially counteracted by an increase in osteoclast resorption activity[45]. Therefore, it is important to explore drugs and protein factors that can simultaneously enhance bone formation and inhibit bone resorption. In conclusion, this study provides new ideas for research on the mechanism underlying osteoblast-osteoclast regulation under inflammatory conditions and provides a potential therapeutic target for clinical treatment.

**Table 1 Primers used for quantitative real-time PCR.**

**Mouse**

| Target | Forward primers, 5′–3′ | Reverse primers, 5′–3′ |
|---|---|---|
| TRAP | AGCAGCCAAGGAGGACTAC | CATAGCCCACACCGTTCTC |
| Cathepsin K | AGCTTCCCCAAGATGTGAT | AGCACCAACGAGAGGAGAA |
| Caltonin receptor | CTCCTTGTCGATTGCTGCT | TCACCCTCTGGCAGCTAAG |
| IL-6 | GCCTTCTTGGGACTGATGCT | GCCATTGCACAACTCTTTTCTCA |
| IL-1β | GTGTCTTTCCCGTGGACCTT | AATGGGAACGTCACACACCA |
| COX-2 | GCATTCTTTGCCCAGCACTT | ACCTCTCCACCAATGACCTGA |
| ALP | CATCGCCTATCAGCTAATGCACA | ATGAGGTCCAGGCCATCCAG |
| Col-1 | GACATGTTCAGCTTTGTGGACCTC | AGGGACCCTTAGGCCATTGTGTA |
| RUNX2 | TCTGACAAAGCCTTCATGTCC | AAATAGTGATACCGTAGATGCG |
| β-action | GGCTGTATTCCCCTCCATCG | CCAGTTGGTAACAATGCCATGT |

## Methods

**Media, reagents, and animals**. Dulbecco's Modified Eagle Medium (DMEM) (catalog no. 8120450) and α-modified Eagle's medium (#8121119) were purchased from Gibco-BRL (Waltham, MA, USA). Specific antibodies against TRAP (#ab191406), CTSK (catalog no. ab187647), and RUNX-2 (#ab236639) were purchased from Abcam (Cambridge, UK); anti-GAPDH (#AF7021), anti-COX-2 (#AF7003), anti-iNOS (#AF0199), anti-TNFR1 (#AF0282), and anti-TNFR2 (#AF0364) from Affinity Biosciences (OH.USA); lamin B1 (#13435), anti-P38 (#8690), anti-p-P38 (#4511), anti-JNK (#9252), anti-p-JNK (#9255), anti-Erk1/2 (#4695), anti-p-Erk1/2 (#8544),anti-AKT (#4691),anti-p-AKT (#4060),anti-IκBα (#4814), anti-p-IκBα (#5209), anti-p-P65 (#3033), and anti-P65 (#8242) from Cell Signaling Technology (Danvers, MA, USA).

We have complied with all relevant ethical regulations for animal use. All experimental animal procedures adhered to institutional guidelines and received approval from the Institutional Animal Care and Use Committee of Shandong University (Shandong, China). Eight to twelve weeks-old female mice were used for experiments.

**Cell culture and treatment**. The murine cell lines RAW264.7 and MC3T3-E1 were derived from the Cell Bank of the Chinese Academy of Sciences. STR analysis was conducted to ensure the authenticity of the cell lines. The references used for authentication included the databases of the Chinese Academy of Sciences and the American Type Culture Collection. MC3T3-E1 cells and primary marrow mesenchymal stem cells were maintained in α-MEM, while RAW264.7 cells and primary bone marrow-derived macrophages were cultured in DMEM. The medium comprised 10% FBS and 1% penicillin/streptomycin. Cells were cultured at 37 °C with 5% $CO_2$ and the culture medium was refreshed every 2 days. All cells were passaged when they reached approximately 80–90% confluence.

Primary bone marrow-derived macrophages (BMDMs) and primary marrow mesenchymal stem cells (BMMSCs)were isolated from mice. After the mice were anesthetized and humanely euthanized by cervical dislocation, they were immersed in disinfectant alcohol for 20 min, followed by the tibia and femur being isolated, and both ends of the epiphyses were cut to the open bone medullary cavity. Flush the marrow into a 15 mL centrifuge tube using a 1 mL syringe. Use 5 mL PBS per bone. Bone marrow cells were collected and seeded in 6-well plates or cell culture dishes. The cells were cultured in a saturated humidity incubator with 5% $CO_2$ at 37 °C. For BMDMs, M-CSF (10 ng/mL, PeproTech, USA) was added to the culture medium for 6 days, and the medium was changed on the third day, BMDMs could be successfully cultured six days later. For BMMSCs, the culture

medium was replaced to remove the unadherent cells after 48 h. The medium was replaced every 3 days. The adherent cells were subcultured after becoming 80–90% confluency. Passages 1–3 cells were used in this study.

**Real-time reverse transcriptase-polymerase chain reaction (RT-PCR)**. Transcriptional expression levels were tested by collecting RAW264.7 and MC3T3-E1 cells. TRIzol reagent (Takara Bio, Japan) was used to extract total RNA, followed by cDNA synthesis using an RT-qPCR kit (Toyota, Japan). SYBR Green PCR Master Mix (Toyobo, Japan) was used to perform the RT-PCR reaction. ΔΔCT was employed to calculate the target gene mRNA expression. Each experiment was replicated three times. The specific primers used for this study included tartrate-resistant acid phosphatase (TRAP), Cathepsin K, Calcitonin Receptor, alkaline phosphatase (ALP), RUNX2, Col-1, Interleukin-6 (IL-6), Interleukin-1β (IL-1β), Cyclooxygenase-2 (COX-2), and β-action. The primer sequences are listed in Table 1.

**Western blotting**. Protein samples were collected from the cell lines and primary cells. Samples containing equal amounts of total protein were separated using 10% SDS-PAGE and then subjected to protein blot analysis. Following electrophoresis, the proteins were transferred onto a polyvinylidene difluoride (PVDF) membrane (Millipore). The PVDF membrane was blocked using 5% skimmed milk and subsequently incubated with a specific primary antibody. An HRP-conjugated secondary antibody was applied for 1 h. Finally, these protein bands were observed using an enhanced chemiluminescence system (Tanon 5200, Tanon, Shanghai, China, and Amersham Imager 600, GE Amersham USA). Particularly, RAW264.7 cells were cultured with TNF-α (10 ng/mL) in the absence or presence of Atsttrin for different time points. Anti-p65 antibody was used for Immunoblotting analysis to test the nuclear extraction (NE) and cytoplasmic extraction (CE). RAW264.7 cells were treated with TNF-α (10 ng/mL) and different concentrations of Atsttrin. Immunoblot analysis was performed with anti-iNOS and anti-COX2 antibodies. The present study found that 500 ng/ml Atsttrin could dramatically enhance osteoblast formation in osteoblastogenesis medium and remarkably inhibit catabolism. In this case, the concentration of 500 ng/ml was used for the in vitro study to make it consistent. The molecular weight of Atsttrin is 16.94 kDa. In this study, the concentration gradient of Atsttrin was set to be 10 ng/ml (0.59 nM), 42.35 ng/ml (2.5 nM), 50 ng/ml (2.95 nM), 100 ng/ml (5.9 nM), 200 ng/ml (11.8 nM), 500 ng/ml (29.5 nM), and 1000 ng/ml (59 nM).

**Enzyme-linked immunosorbent assay**. BMDMs were inoculated into 6-well cell culture plates, and the BMDM culture medium

was supplemented with Macrophage Colony-Stimulating Factor (M-CSF) (10 ng/mL) (PeproTech, USA) for 6 days. Subsequently, the BMDMs were treated without or with TNF-α (10 ng/mL) (PeproTech, USA) in the absence or presence of Atsttrin (500 ng/ml) for 48 h. The levels of IL-1β and IL-6 in the supernatant were assessed by utilizing the ELISA kit (Elabscience, USA) following the provided guidelines from the manufacturer. Moreover, the levels of CTSK and TRAP in the mouse serum were assayed through ELISA using a commercial kit (Elabscience, USA) as per the manufacturer's guidelines.

**Osteoclastogenesis**. BMDMs and RAW264.7 cells were treated with TNF-α (10 ng/ml), RANKL (PeproTech, USA, 100 ng/ml), and Atsttrin (500 ng/ml) for 7 days. The culture medium was refreshed every 2 days.

**Tartrate-resistant acid phosphatase (TRAP) staining**. The cells were fixed in 4% paraformaldehyde for 20 min. The next day, cells were stained with a TRAP Assay Kit (G1492; Beijing Solarbio). All procedures strictly followed the manufacturer's instructions. Osteoclasts were identified as TRAP-positive cells with three or more nuclei.

**Osteoblastogenesis**. BMMSCs were isolated by the above methods. Then, the cells were plated in a six-well plate and incubated in α-MEM, which was supplemented with 1% penicillin/streptomycin and 10% FBS. One day later, the cells were rinsed with PBS and then exposed to osteoblastogenesis medium, consisting of basal medium with 10% FBS, 10 nM dexamethasone, 5 mM β-glycerophosphate, and 50 μg/ml ascorbate-2-phosphate. The osteoblastogenesis medium was regularly replaced every three days to ensure optimal conditions for cell growth and differentiation.

**Alkaline phosphatase (ALP) staining**. BMMSCs were nurtured in the osteoblastogenesis medium while being exposed to 10 ng/ml TNF-α with or without Atsttrin (500 ng/ml) for 7 days. Cells were treated with BCIP/NBT Alkaline Phosphatase Color Development Kit (C3206; Beyotime Biotechnology).

**Alkaline phosphatase assay**. Cell lysis buffer without inhibitors (P0013J; Beyotime Biotechnology) was used to lyse the samples. Subsequently, the cell lysate was gathered and the protein concentration was evaluated employing the BCA assay kit (P0012; Beyotime Biotechnology). The samples were determined using the ALP assay kit (P0321; Beyotime Biotechnology). The measurement of the absorbance optical density (OD) took place at a wavelength of 405 nm using the xMark Microplate Absorbance Spectrophotometer (Bio-Rad).

**Knockdown of TNFR1 and TNFR2 by siRNA**. The cells were transfected with siTNFR1 and siTNFR2 and scrambled control siRNA (scRNAi) using lipofectamine 2000 (11668-019, Invitrogen), according to the instructions. After 24 h, the culture medium changed. Western blotting was employed to assess the effectiveness of siRNA transfection. Lysate samples were gathered and subjected to mRNA and protein analysis after varying treatments.

**CCK8 assay**. To examine the effect of Atsttrin (500 ng/ml) on cell proliferation, Cell viability was assessed using the Cell Counting Kit-8 (CCK8, Beyotime, Shanghai, China). The procedures were executed following the guidelines provided by the manufacturer.

**Ovariectomy-induced osteoporosis animal model**. Female C57BL/6 mice, aged eight weeks, were anesthetized with 1% sodium pentobarbital and subjected to sham operation or bilateral ovariectomy. After 8 weeks, the mice were intraperitoneally injected with PBS or Atsttrin (2.5 mg/kg body weight) twice per week for 8 weeks. Then, the femurs were harvested for analysis.

**Micro-CT scanning**. The femur tissues were fixed overnight in 70% ethanol, after which they underwent scanning using an in vivo micro-CT imaging system specially designed for small animals (QuantumGX2, Massachusetts, USA). Scan parameters were set at 6 μm per pixel resolution, 88 mA current, and 90 kV voltage. Two hundred section planes below the growth plate of the distal metaphysis as the volume of interest (VOI) was used for reconstruction and analysis[46]. Further, we constructed a three-dimensional model and analyzed bone mineral density (BMD), bone volume/tissue volume (BV/TV), and trabecular number (Tb.N).

**Statistics and reproducibility**. All data are presented as the mean ± standard deviation. All statistical analyses were analyzed using the GraphPad Prism software (version 7.0; GraphPad Inc., La Jolla, CA, USA). Student $t$-tests were used for comparisons between two independent groups, and one-way or two-way ANOVA (when appropriate) was performed. The experiments were conducted independently at least three times with duplicate samples. Statistical significance at two sides was indicated when $P < 0.05$.

**Reporting summary**. Further information on research design is available in the Nature Portfolio Reporting Summary linked to this article.

## Data availability

The datasets used and/or analyzed in this study can be found in Supplementary Material. Original blots are provided in Supplementary Material and the original date source is provided in Supplementary Data 1. Unedited blots of the blots were shown in Supplementary Figs. 2–9. Further information and requests for resources and reagents are available from the corresponding author.

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

## Acknowledgements

This work was supported partly by the National Key Research and Development Program of China (2020YFC2009004), the National Natural Science Foundation of China (81900804), the Natural Science Foundation of Shandong Province (ZR2019BH071), and (2019GSF108029), and China Postdoctoral Science Foundation Grant (2019M651064).

Ethics Approval: This study was approved by medical ethics regulations of the Medical Ethical Committee of Qilu Hospital of Shandong University.

## Author contributions

J.W. and L.C. designed the experiments; K.L. and Z.W. acquired the data, J.L., W.Z., F.Q., and Z.W. performed the statistical analyses; Q.H. analyzed and interpreted the data; J.S. and Q.M. maintained the mice; J.W. and K.L. edited the paper. All authors drafted and reviewed the paper.

## Competing interests

The authors declare no competing interests.
