## [Peer Review File · Communications Biology]

Reviewers' comments:

Reviewer #1 (Remarks to the Author):

Overall, this manuscript has potential, but especially the correct use of English should be improved considerably.

- Line 19: "Present study 19 use cell TRAP staining to determine the role of Atsttrin in osteoclastogenesis." This is not correct. TRAP staining is usually used to stain the presence of osteoclasts or to monitor whether osteoclastogenesis has taken place.
- Line 23-24: What do the authors mean by inflammatory destruction? How was this measured?
- Line 24: Mechanistic rather than mechanism.
- It is not clear in the abstract based on what experiments they concluded a role for TNFR1 rather than TNFR2.
- Line 34: the disorder of cytokines... Not proper English.
- Line 40: Also: ... by making ... is not proper English.
- PGRN rather full name.
- Line 110: "Absorb the supernatant and determine the protein concentration ..." is not proper English.
- Line 181: a predominant role.
- Line 185: Otherwise Is not a correct English phrase.

From line 185 I started to ignore the English, which should be improved tremendously before resubmission.

In figure 1a: we see the osteoclast paradox: more large multinucleated cells when RANKL and TNF are added simultaneously. This should go to the expense of the total number of multinucleated cells, since smaller ones are no longer there. So, what is seen in Fig. 1b? Mutlinucleated TRAP+ cells?

I am quite surprised that osteoclasts express IL-17 (fig 2) , this is in particular or exclusively a cytokine expressed in T-cells. How reliable is this? Given the expected extremely low expression, I would suggest that all expression data are NOT normalized (control =1), but rather display relative expression divided by the housekeeping gene expression. If done as such, one can compare the level of for instance TNF or IL-1beta with IL-17 and judge that it is much lower. Line 490: Inflammatory catabolism seems double.

In Fig. 4A: there are hardly any cells visible in the second panel. Thus, I am not sure whether the conditions are comparable.

Figure 8 shows a translation in a mouse model, where Atsttrin shows an effect on bone parameters.

Reviewer #2 (Remarks to the Author):

The authors of the manuscript „ Atsttrin regulates osteoblastogenesis and osteoclastogenesis through TNFR pathway” investigated the role of the PRGN-derived synthetic protein Atsttrin in inflammation-associated osteoporosis. Therefore, they analyzed the effects of Atsttrin on TNF- α stimulated osteoclast or osteoblast formation by respective cell lines. Their main findings were that Atsttrin prevented TNF- α effects via TNFR1 and further promoted osteoblastogenesis through TNFR2 signaling.

The authors further provided first results in an OVX osteoporosis mouse model.

The finding and investigation of Atsttrin as a treatment option in inflammation-related bone diseases such as osteoporosis is valuable, but regarding the manuscript data interpretation and novelty, I have some concerns:

Major:

1. Atsttrin as its origin molecule PRGN is an antagonist of TNF- α signaling, thus, it is obvious that adding Atsttrin to TNF- α stimulation will inhibit the inflammatory effects in general. Here, this is shown for osteoclastogenesis, immune activation and osteoblast formation. A competitive mechanism is assumable which is depending on binding affinity and concentration, but Atsttrin (500-1000 ng/ml) concentration is >20 fold higher than TNF- α (10 ng/ml) which might cause a bias... The authors should provide molarities for a better comparison.

2. It is not surprising that there is no effect of TNF- α after silencing of TNFR1. It can not be stated that this proves that Atsttrin is mediating its effects through TNFR1 in a double stimulated approach. This has to be compared to TNFR1 knock down on TNF- α or Atsttrin alone.

3. The authors show that Atsttrin promotes osteoblastogenesis via TNFR2-Akt-Erk1/2 signaling. But as they already showed this for chondrogenesis (PMID: 33195216), the novelty is rather limited.

Minor:

4. Pictures and quantifications are not fitting: e.g. in Fig. 1A there are almost no osteoclasts in the RANKL+TNF- α +Atsttrin group, but in 1B numbers are as high as for RANKL control. Same for 3F/G.

This also occurs for WB: e.g. Fig. 1B band densities are not fitting to 1D.

5. 8 hours is a quiet early time point to measure gene expression induction of late osteoclast markers such as TRAP or CatK.

6. RANKL+Atsttrin control should be provided (Fig. 1A), TNFR1 silencing is only shown for double treated group, but should be also compared to Atsttrin or TNF- α separately (Fig. 3/ Fig. 5), basal expression of the genes by unstimulated cells have to be included to proof success of differentiation.

7. Information on the VOI for μ CT analysis has to be provided. OVX femur looks overall smaller and BMD is unusual low. Why are only 3 data point shown: 3 mice*2 legs?

Overall, the English has to be improved. There are wrongly nested sentences (e.g. lines 44-49), misspellings and grammar mistakes throughout the manuscript.

Atsttrin has the potential as a therapeutic in inflammatory-related bone diseases and it is important to investigate its effects in different settings. But as it was already shown by the authors that Atsttrin inhibits TNF- α signaling via TNFR1 in an arthritis model (PMID: 29258611) and exerts supportive effects on chondrogenic differentiation via TNFR2 signaling (PMID: 33195216), the here presented data on osteoclast formation and osteoblastogenesis are more of an additive information content. Because of this and the above-mentioned concerns I cannot recommend this manuscript for publication in Communications Biology.

We greatly appreciate the opportunity to publish our study. Many thanks for the reviewers' work, which may have improved the quality of our manuscript. The following is a point-by-point response.

Reviewers' comments:

Reviewer #1

Overall, this manuscript has potential, but especially the correct use of English should be improved considerably.

- Line 19: "Present study use cell TRAP staining to determine the role of Atsttrin in osteoclastogenesis." This is not correct. TRAP staining is usually used to stain the presence of osteoclasts or to monitor whether osteoclastogenesis has taken place.

R: We thank the reviewer for the suggestion. We replaced this sentence with "The present study used cell TRAP staining to determine whether Atsttrin affects osteoclast formation".

- Line 23-24: What do the authors mean by inflammatory destruction? How was this measured?

R: Thank you for the reviewer's suggestion. We modified this sentence instead of "we found that Atsttrin inhibited TNF- α -induced osteoclastogenesis and inflammation". We measured the inflammatory molecules in Figure 2A-I.

- Line 24: Mechanistic rather than mechamism.

R: We corrected this grammatical mistake.

- It is not clear in the abstract based on what experiments they concluded a role for TNFR1 rather than TNFR2.

R: Thank you for the reviewer's suggestion. We briefly stated this in the abstract. We used TNFR1 or TNFR2 RNA silencing technology to downregulate TNFR expression. By blocking the pathway, we found that Atsttrin exhibited its effect through both TNFR1 and TNFR2 signaling.

- Line 34: the disorder of cytokines... Not proper English.

R: We corrected this word to "inflammatory cytokines".

- Line 40: Also: ... by making ... is not proper English.

R: We replaced the words with "in".

- PGRN rather full name.

R: We described the full name of PGRN instead of a short name.

- Line 110: "Absorb the supernatant and determine the protein concentration ..."
is not proper English.

R: we modified this sentence" supernatant was collected and protein concentration was determined by BCA protein detection kit"

- Line 181: a predominant role.

R: We corrected this mistake.

- Line 185: Otherwise ... Is not a correct English phrase.

R: We corrected this phrase as suggested.

From line 185 I started to ignore the English, which should be improved tremendously before resubmission.

In figure 1a: we see the osteoclast paradox: more large multinucleated cells when RANKL and TNF are added simultaneously. This should go to the expense of the total number of multinucleated cells, since smaller ones are no longer there. So, what is seen in Fig. 1b? Mutlinucleated TRAP+ cells?

R: The manuscript was edited by *American Journal Experts* as the reviewer suggested. We defined osteoclasts as those with three or more nuclei (PMID: 33208358; PMID: 36966936). In this panel, we counted the osteoclast number. We corrected the label accordingly.

I am quite surprised that osteoclasts express IL-17 (fig 2) , this is in particular or exclusively a cytokine expressed in T-cells. How reliable is this? Given the expected extremely low expression, I would suggest that all expression data are NOT normalized (control =1), but rather display relative expression divided by the housekeeping gene expression. If done as such, one can compare the level of for instance TNF or IL-1beta with IL-17 and judge that it is much lower.

Line 490: Inflammatory catabolism seems double.

R: Thank you for your suggestion. Classically, IL-17 is mainly released by T cells. Additionally, recent studies have reported that BMDMs also release IL-17 (PMID: 31231373; PMID: 30055802; PMID: 28871251). Indeed, we displayed relative expression divided by housekeeping gene expression. To avoid this possible misunderstanding, we removed IL-17 determination in the revised manuscript.

In Fig. 4A: there are hardly any cells visible in the second panel. Thus, I am

not sure whether the conditions are comparable.

R: We thank the reviewer for the suggestion. We reperformed ALP staining and performed quantitative analysis.

Figure 8 shows a translation in a mouse model, where Atsttrin shows an effect on bone parameters.

R: We thank the reviewer for the suggestion. We established an OVX mouse model and performed micro-CT scanning. As indicated in Figure 8A&B, Atsttrin significantly improved surgically induced osteoporosis. Moreover, the relative bone parameters, such as BV/TV and Tb.N, were remarkably rescued by the additional use of Atsttrin (Figure 8C&D).

Reviewer #2 (Remarks to the Author):

The authors of the manuscript “Atsttrin regulates osteoblastogenesis and osteoclastogenesis through TNFR pathway” investigated the role of the PRGN-derived synthetic protein Atsttrin in inflammation-associated osteoporosis. Therefore, they analyzed the effects of Atsttrin on TNF- α stimulated osteoclast or osteoblast formation by respective cell lines. Their main findings were that Atsttrin prevented TNF- α effects via TNFR1 and further promoted osteoblastogenesis through TNFR2 signaling. The authors further provided first results in an OVX osteoporosis mouse model.

The finding and investigation of Atsttrin as a treatment option in inflammation-related bone diseases such as osteoporosis is valuable, but regarding the manuscript data interpretation and novelty, I have some concerns:

Major:

1. Atsttrin as its origin molecule PRGN is an antagonist of TNF- α signaling, thus, it is obvious that adding Atsttrin to TNF- α stimulation will inhibit the inflammatory effects in general. Here, this is shown for osteoclastogenesis, immune activation and osteoblast formation. A competitive mechanism is assumable which is depending on binding affinity and concentration, but Atsttrin (500-1000 ng/ml) concentration is >20 fold higher than TNF- α (10 ng/ml) which might cause a bias... The authors should provide molarities for a better comparison.

R: Thanks for the reviewer's suggestion. We provided the molarities of Atsttrin and TNF- α . As indicated in the Supplemental Figure legends A, B & C, Atsttrin inhibited TNF- α in a dose-dependent manner.

2. It is not surprising that there is no effect of TNF- α after silencing of TNFR1. It can not be stated that this proves that Atsttrin is mediating its

effects through TNFR1 in a double stimulated approach. This has to be compared to TNFR1 knock down on TNF- α or Atsttrin alone.

R: Thanks for the reviewer' s suggestion. We regrouped the data as follows:

Fig. 3B-D: a) blank group; b) scRNAi ,RANKL; c) scRNAi,RANKL,TNF- α ; d) scRNAi,RANKL, TNF- α , Atsttrin; e) TNFR1 RNAi, RANKL, TNF- α ; f) TNFR1 RNAi, RANKL, Atsttrin; g) TNFR1 RNAi, RANKL, TNF- α , Atsttrin.

Fig. 3E-I: a) scRNAi , RANKL; b) scRNAi, RANKL, TNF- α ; c) scRNAi, RANKL, TNF- α , Atsttrin; d) TNFR1 RNAi, RANKL, TNF- α ; e) TNFR1 RNAi, RANKL, Atsttrin; f) TNFR1 RNAi, RANKL, TNF- α , Atsttrin.

Fig. 5B-H: a) scRNAi; b) scRNAi, TNF- α ; c) scRNAi, TNF- α , Atsttrin; d) TNFR1 RNAi, TNF- α ; e) TNFR1 RNAi, Atsttrin; f) TNFR1 RNAi, TNF- α , Atsttrin.

As illustrated in Figure 3B-I, Atsttrin inhibited TNF- α -induced osteoclastogenesis via TNFR1. Additionally, as shown in Figure 5B-H, Atsttrin rescued TNF- α -mediated inhibition of osteoblastogenesis through TNFR1.

3. The authors show that Atsttrin promotes osteoblastogenesis via TNFR2-Akt-Erk1/2 signaling. But as they already showed this for chondrogenesis (PMID: 33195216), the novelty is rather limited.

R: Thank you for your suggestion. A previous study found that Atsttrin promoted chondrogenesis via TNFR2 signaling. In the present study, we determined Atsttrin' s function in different conditions with different pathogenesis. Given Atsttrin' s nature, we investigated and confirmed the molecular mechanism involved in osteoporosis. Even though Atsttrin exerted its promotive effect through the classic pathway, the present study illustrated a novel potential alternative for Atsttrin.

Minor:

4. Pictures and quantifications are not fitting: e.g. in Fig. 1A there are almost no osteoclasts in the RANKL+TNF- α +Atsttrin group, but in 1B numbers are as high as for RANKL control. Same for 3F/G. This also occurs for WB: e.g. Fig. 1B band densities are not fitting to 1D.

R: Thanks for the reviewer' s suggestion. We replaced the low-quality figure as mentioned. (Figure 1A&B, Figure 3E&F).

Figure 1D shows the relative band density based on Figure 1C. The present study found that RANKL induced TRAP expression and that TNF- α enhanced TRAP expression in the presence of RANKL. Interestingly, additional use of atsttrin reduced TNF- α ' s effect.

5. 8 hours is a quiet early time point to measure gene expression induction of late osteoclast markers such as TRAP or CatK.

R: We thank the reviewer for the suggestion. Although some studies indicated that 8 hours was sufficient for TRAP gene determination (PMID: 23942871), we redetermined the transcriptional expression of these genes after 24 hours of treatment (Figure 1F-H; Figure 3G-I). TNF- α prompted robust osteoclastogenesis

by osteoclast precursors pretreated with RANKL (PMID: 11032840; 11500955).

6. RANKL+Atsttrin control should be provided (Fig. 1A), TNFR1 silencing is only shown for double treated group, but should be also compared to Atsttrin or TNF- α separately (Fig. 3/ Fig. 5), basal expression of the genes by unstimulated cells have to be included to proof success of differentiation.

R: Thanks for the reviewer' s suggestion. We regrouped Figure 1A&C and Fig. 3/Fig. 5 as follows:

Figure 1A:

a) RANKL; b) RANKL+Atsttrin; c) RANKL+TNF α ; d) RANKL+TNF α +Atsttrin; e) TNF α ; f) TNF α +Atsttrin

Figure 1C:

a) blank group; b) RANKL; c) RANKL+Atsttrin; d) RANKL+TNF α ; e) RANKL+TNF α +Atsttrin

As illustrated in 1A&C, Atsttrin inhibited TNF α -induced osteoclastogenesis. Additionally, Atsttrin did not inhibit RANKL-induced osteoclast differentiation in vitro.

Fig. 3B-D:

a) blank group; b) scRNAi , RANKL; c) scRNAi, RANKL, TNF- α ; d) scRNAi, RANKL, TNF- α , Atsttrin; e) TNFR1 RNAi, RANKL, TNF- α ; f) TNFR1 RNAi, RANKL, Atsttrin; g) TNFR1 RNAi, RANKL, TNF- α , Atsttrin.

Fig. 3E-I:

a) scRNAi , RANKL; b) scRNAi, RANKL, TNF- α ; c) scRNAi, RANKL, TNF- α , Atsttrin; d) TNFR1 RNAi, RANKL, TNF- α ; e) TNFR1 RNAi, RANKL, Atsttrin; f) TNFR1 RNAi, RANKL, TNF- α , Atsttrin.

Fig. 5B-H:

a) scRNAi; b) scRNAi, TNF- α ; c) scRNAi, TNF- α , Atsttrin; d) TNFR1 RNAi, TNF- α ; e) TNFR1 RNAi, Atsttrin; f) TNFR1 RNAi, TNF- α , Atsttrin.

As illustrated in Figure 3B-I, Atsttrin inhibited TNF- α -induced osteoclastogenesis via TNFR1. Additionally, as shown in Figure 5B-H, Atsttrin rescued TNF- α -mediated inhibition of osteoblastogenesis through TNFR1.

7. Information on the VOI for μ CT analysis has to be provided. OVX femur looks overall smaller and BMD is unusual low. Why are only 3 data point shown: 3 mice*2 legs?

R: Thank you for your suggestion. Two hundred section planes below the growth plate of the distal metaphysis as the volume of interest (VOI) were used for reconstruction and analysis. The present study used 6 mice per group, and we updated the data.

Overall, the English has to be improved. There are wrongly nested sentences (e.g. lines 44-49), misspellings and grammar mistakes throughout the manuscript.

R: The manuscript was edited by *American Journal Experts* as the reviewer

suggested.

Atsttrin has the potential as a therapeutic in inflammatory-related bone diseases and it is important to investigate its effects in different settings. But as it was already shown by the authors that Atsttrin inhibits TNF- α signaling via TNFR1 in an arthritis model (PMID: 29258611) and exerts supportive effects on chondrogenic differentiation via TNFR2 signaling (PMID: 33195216), the here presented data on osteoclast formation and osteoblastogenesis are more of an additive information content. Because of this and the above-mentioned concerns I cannot recommend this manuscript for publication in Communications Biology.

Reviewers' comments:

Reviewer #1 (Remarks to the Author):

My comments were appropriately answered. However, I reread the manuscript and here are a few old/new ones:

In abstract, I would suggest to replace the sentence "The present study used cell TRAP staining to determine

whether Atsttrin affects osteoclast formation" by: " The present study investigated whether Atsttrin affected osteoclast formation". The way it is written now, is very confusing: in essence, one can use TRAP staining to visualize multinucleated and mononuclear cells.

Minor:

line 51 After progranulin add a space before (. Likewise in line 134, a space should be inserted before (.

Fig 1H it should be Calcitonin Receptor, not Caltonin. Likewise in Fig 3 I

Reviewer #2 (Remarks to the Author):

I appreciate that the authors improved their manuscript and provided more information. However, for me it is still not clear why the authors used such a high Atsttrin concentration (500-1000 ng/ml) compared to the initial study describing Atsttrin function (29.5-59 nM vs. 2.5 nM, PMID: 21393509) and also compared to associated papers (200 ng/ml were sufficient to show effects, PMID: 29258611). Does this high concentration have side effects on viability and function of cells? CCK8 assay is mentioned in the methods but results are missing...

Microscopic pictures show only limited osteoclast formation in the RANKL treated samples and "true" osteoclasts seemed to have only formed in the samples additionally treated with TNF- α . Further, unstimulated controls that are mandatory to evaluate success of differentiation are missing in the gene expression analysis and densitometric analysis of WB is not always fitting to the band appearance in the shown blots.

Additionally, the effects of Atsttrin on osteoclastogenesis were previously described as a measure for anti-TNF α activity of Atsttrin (PMID: 33185871, here 100 ng/ml Atsttrin was sufficient) and at least this has to be mentioned in the manuscript.

In the animal model and mCT analysis, was there now 3 mice used with 6 legs evaluated or 6 mice and then only 1 leg evaluated. If so, why? The information given are not consistent.

Unfortunately, the abstract does not provide any conclusion and the link of the in vitro data to osteoporosis and the data from the OVX model is missing.

I agree with the authors that finding new therapeutic options for osteoporosis is important. The authors provide a lot of in vitro data regarding the effect of Atsttrin on osteoclastogenesis as well as on osteoblastogenesis and preliminarily evaluated Atsttrin in an osteoporosis mouse model. With this study the authors confirm the known functions of Atsttrin (TNF- α agonist via TNFR1 and pro-regenerative effects via TNFR2-Akt and/or ERK1/2 signaling) in another in vitro setting and provided ideas for a potential application of PGRN/Atsttrin in osteoporosis treatment.

Overall, the amount of data is somehow overwhelming with different aspects being addressed and put together within one manuscript. I have to admit that I am still not convinced regarding the quality of

the experiments and originality of the findings, and thus, I still have concerns to recommend this study for publication in *Communications Biology*.

Much appreciated for the opportunity to improve our work. We addressed the reviewers' comments, particularly, we clearly stated the reason for Atsttrin's concentration and provided the control group as required. Following is the point-to-point response.

Reviewer #1:

My comments were appropriately answered. However, I reread the manuscript and here are a few old/new ones:

In abstract, I would suggest to replace the sentence "The present study used cell TRAP staining to determine whether Atsttrin affects osteoclast formation" by: " The present study investigated whether Atsttrin affected osteoclast formation". The way it is written now, is very confusing: in essence, one can use TRAP staining to visualize multinucleated and mononuclear cells.

R: We thank the reviewer for the suggestion. We replaced this sentence with "The present study investigated whether Atsttrin affected osteoclast formation".

Minor:

line 51 After progranulin add a space before (. Likewise in line 134, a space should be inserted before (.

Fig 1H it should be Calcitonin Receptor, not Caltonin. Likewise in Fig 3 I

R: We thank the reviewer for the suggestion. We corrected these mistakes.

Reviewer #2:

I appreciate that the authors improved their manuscript and provided more information. However, for me it is still not clear why the authors used such a high Atsttrin concentration (500-1000 ng/ml) compared to the initial study describing Atsttrin function (29.5-59 nM vs. 2.5 nM, PMID: 21393509) and also compared to associated papers (200 ng/ml were sufficient to show effects, PMID: 29258611). Does this high concentration have side effects on viability and function of cells? CCK8 assay is mentioned in the methods but results are missing...

R: Thanks for the reviewer's suggestion. To address the issue of Atsttrin's concentration effect on osteoclast and osteoblast formation, we established an in vitro cell model with various concentrations. As indicated in Figures A&B&C, with concentrations of 42.35 ng/ml (2.5 nM) and 50 ng/ml (2.95 nM), Atsttrin exerted sufficient anti-TNF α 's effect. In addition, Atsttrin showed anti-catabolic effect in a dose-dependent manner with concentrations of 100 ng/ml (5.9 nM), 200 ng/ml (11.9 nM), and 500 ng/ml (29.5 nM), respectively. Especially, with the concentration of 1000 ng/ml (59 nM), Atsttrin almost abolished TNF α 's effect. The concentration of 2.5 nM was the same as the previous publication (PMID: 21393509).

To determine the concentration of Atsttrin for activating osteoblast formation, the present study took advantage of 200 ng/ml (11.9 nM) and 500 ng/ml (29.5 nM) Atsttrin. As illustrated in Figure S D, 200 ng/ml (11.9 nM) Atsttrin weakly activates Erk1/2 or Akt pathway, while 500 ng/ml (29.5 nM) Atsttrin could strongly activate Erk1/2 and Akt pathway. To make the

concentration consistent in the whole manuscript, the present study took a concentration of 500ng/ml(29.5nM) to investigate the role of Atsttrin in osteoblast and osteoclast formation. The reason for Atsttrin concentration determination was stated in the M&M part as required. The details of the molarity calculation for Atsttrin were provided in the M&M part. The reason for choosing 500ng/ml was also discussed in the Discussion part. To test whether this concentration affected cell viability, the present study used a CCK-8 assay. As demonstrated in Figure S J&K, 500ng/ml(29.5nM) Atsttrin could not affect cell viability.

Microscopic pictures show only limited osteoclast formation in the RANKL treated samples and "true" osteoclasts seemed to have only formed in the samples additionally treated with TNF- α . Further, unstimulated controls that are mandatory to evaluate success of differentiation are missing in the gene expression analysis and densitometric analysis of WB is not always fitting to the band appearance in the shown blots.

R: Thank you for the reviewer's suggestion. We regrouped the data as follows:

Fig. 1: a) blank; b) RANKL+Atsttrin; c) RANKL+TNF- α ; d) RANKL+TNF- α +Atsttrin; e) TNF- α ; f) TNF- α +Atsttrin.

Fig. 3: a) blank; b) scRNAi, RANKL; c) scRNAi, RANKL, TNF- α ; d) scRNAi, RANKL, TNF- α , Atsttrin; e) TNFR1 RNAi, RANKL, TNF- α ; f) TNFR1 RNAi, RANKL, Atsttrin; g) TNFR1 RNAi, RANKL, TNF- α , Atsttrin.

In this case, the present study included the unstimulated controls as required. Additionally, the transcriptional expression was analyzed as well as WB band density.

Additionally, the effects of Atsttrin on osteoclastogenesis were previously described as a measure for anti-TNF α activity of Atsttrin (PMID: 33185871, here 100 ng/ml Atsttrin was sufficient) and at least this has to be mentioned in the manuscript.

R: Thank you for the reviewer's suggestion. We have added relevant content to the discussion section of the manuscript.

In the animal model and mCT analysis, was there now 3 mice used with 6 legs evaluated or 6 mice and then only 1 leg evaluated. If so, why? The information given are not consistent.

R: We thank the reviewer for the suggestion. In the present study, we used 6 mice per group for the study. 1 leg was evaluated for each mouse. We have corrected the mislabeling in Figure Legends 8.

Unfortunately, the abstract does not provide any conclusion and the link of the in vitro data to osteoporosis and the data from the OVX model is missing.

R: We thank the reviewer for the suggestion. We have added a relevant description "Further, recombinant Atsttrin protected against bone loss in the ovariectomy-induced osteoporosis mouse model *in vivo*" .

REVIEWERS' COMMENTS:

Reviewer #2 (Remarks to the Author):

None